

# Mathematical models are a powerful method to understand and control the spread of Huanglongbing

Rachel A. Taylor[1], Erin A. Mordecai[2], Christopher A. Gilligan[3], Jason R. Rohr[1] and Leah R. Johnson[1,4]

[1] Department of Integrative Biology, University of South Florida, Tampa, Florida, United States
[2] Department of Biology, Stanford University, Stanford, California, United States
[3] Department of Plant Sciences, University of Cambridge, Cambridge, United Kingdom
[4] Department of Statistics, Virginia Polytechnic Institute and State University (Virginia Tech), Blacksburg, Virginia, United States

## ABSTRACT

Huanglongbing (HLB), or citrus greening, is a global citrus disease occurring in almost all citrus growing regions. It causes substantial economic burdens to individual growers, citrus industries and governments. Successful management strategies to reduce disease burden are desperately needed but with so many possible interventions and combinations thereof it is difficult to know which are worthwhile or cost-effective. We review how mathematical models have yielded useful insights into controlling disease spread for other vector-borne plant diseases, and the small number of mathematical models of HLB. We adapt a malaria model to HLB, by including temperature-dependent psyllid traits, "flushing" of trees, and economic costs, to show how models can be used to highlight the parameters that require more data collection or that should be targeted for intervention. We analyze the most common intervention strategy, insecticide spraying, to determine the most cost-effective spraying strategy. We find that fecundity and feeding rate of the vector require more experimental data collection, for wider temperatures ranges. Also, the best strategy for insecticide intervention is to spray for more days rather than pay extra for a more efficient spray. We conclude that mathematical models are able to provide useful recommendations for managing HLB spread.

Corresponding author
Rachel A. Taylor,
rataylor21@gmail.com

## INTRODUCTION

Huanglongbing (HLB), also known as citrus greening disease, is a devastating citrus disease native to Asia (*Bové, 2006*; *Gottwald, 2010*; *Hall et al., 2013*) that now exists in virtually all citrus-growing regions (*Narouei-Khandan et al., 2016*). In the last 10 years, it invaded the Western Hemisphere, primarily Brazil and Florida, where it has spread rapidly and caused extensive economic burdens (*Hodges & Spreen, 2012*; *Spreen et al., 2006*). HLB is caused by three bacteria: *Candidatus* Liberibacter asiaticus (CLas), *Candidatus* Liberibacter africanus, and *Candidatus* Liberibacter americanus. The Asian citrus psyllid (ACP), *Diaphorina citri* Kuwayama, is the primary vector

(*Grafton-Cardwell, Stelinski & Stansly, 2013*). HLB causes chlorosis of leaves, dieback and, in severe cases, tree death. Additionally, infected trees develop fruit that is of poor quality and drops early, reducing yields of edible and marketable fruit from diseased trees (*Bové, 2006*). HLB is undermining the viability of an important international industry and possibly endangering the persistence of multiple species of citrus (*Hall et al., 2013*).

## Intervention strategies for citrus greening

Nowhere in the world is citrus greening under adequate control (*Gottwald, 2010*; *Hall et al., 2013*). The process of finding effective intervention strategies has been challenging, at least partly because of the difficulties in determining the infection status of trees and the long duration before trees show symptoms (*Manjunath et al., 2008*; *Gottwald, 2010*). The current state of control involves insecticide spraying to reduce the abundance of ACP (*Grafton-Cardwell, Stelinski & Stansly, 2013*).

To fight citrus greening disease, new intervention strategies are needed. This could be by developing new controls or by combining current and new controls into an optimal strategy (*Halbert & Manjunath, 2004*; *Wang & Trivedi, 2013*). However, before controls can be implemented in the field they need to be tested for efficacy. There are presently tens if not hundreds of hypothetical interventions that could be tested, including: antibiotics (*Zhang et al., 2014*), pesticides (*Qureshi, Kostyk & Stansly, 2014*), biocontrol agents (*Michaud, 2002*), heat treatment (*Hoffman et al., 2013*), new tolerant or resistant tree stocks (*Dutt et al., 2015*), nutrient additions (*Gottwald et al., 2012*), tree removal (*Gottwald, 2010*), changes to tree spacing (*Martini, Pelz-Stelinski & Stelinski, 2015*), intercropping (*Gottwald et al., 2014*), and psyllid deterrents and barriers (*Tisgratog et al., 2016*; *Tomaseto, Krugner & Lopes, 2016*). Even more daunting are the different factorial combinations of interventions to test. It would be difficult and costly to test this large number of potential intervention methods, as well as combinations of these, in the field. Instead, it would be better to start first with those that have the most potential, both in terms of success at reducing the rate or severity of the disease and the costs for implementing the strategy. The question is how to identify these strategies.

Here, we argue that collaborations between empiricists and mathematical modelers can more efficiently identify effective control strategies for HLB. There is a long history of mathematical models of other vector-borne diseases being used to quickly and reliably identify the parameters of the host-vector-pathogen system that are most sensitive to perturbations and thus controls. By coupling these models with models of the economics costs of various interventions, combined cost-benefit analyses can quickly and reliably guide the formidable task of empirically testing HLB interventions. Indeed, mathematical models can provide insights into the cost effectiveness of lone and combined intervention strategies faster than almost any other approach. They can help efficiently target experiments and field data collection on particular critical factors and interventions. The empirical data can then in turn be used to test, validate, and refine the models. Thus, by combining appropriate models with laboratory and field experiments, we expect to develop more cost-effective interventions more quickly than using empirical approaches alone.

## The usefulness of mathematical models

Mathematical models for disease systems were first developed by *Kermack & McKendrick (1927)*, which paved the way for many future models. In these models, individuals move between different compartments depending on their disease status—often "Susceptible," "Infected" and "Recovered," and thus the models are referred to as SIR models. More detailed versions of these models have since evolved to include elements such as demography of the population, age structure, exposure periods, asymptomatic individuals, waning immunity and vector-borne transmission. *SIR* models for vector-borne diseases were initially developed by *Ross (1910)*, *Macdonald (1952)* and *Macdonald (1961)*. The purpose of these mathematical models falls along a continuum between "strategic" or "tactical" models (*Nisbet & Gurney, 1982*). With strategic models, the question the modelers wish to answer is "What could possibly happen?" They aim to find general conclusions that can be used to understand the drivers of population change across many systems. The models are often poor representations of real data. Tactical models, in contrast, are inherently connected to a system and to data collected. Their focus is to make predictions but their answers are only applicable to that one system and are not easily generalized. Furthermore, they usually are unable to show why things occur as they give no information on the drivers of the system. By connecting strategic models more closely to data, it is possible to make qualitative predictions and yet retain understanding of what are the important elements of the system—so that it is possible to understand the effects of targeting specific parameters for control.

One method for ensuring that the qualitative predictions are sensible is through sensitivity analysis—analyzing the importance of different parameters on key disease measures. Sensitivity analysis can alert us to cases when we need more data to be sure of our predictions. It also highlights which parameters are best to change if we want to affect some aspect of the system, such as the timing or size of disease outbreaks. Further we can incorporate information about costs of changing parameters to a strategic model and attempt to optimize the solution. That is, it allows us to be able to choose, based on some measure of profit, which management strategy out of many is the best in terms of maximizing outcomes while minimizing cost. Mathematical models are able to consider many possible intervention strategies, compare them cost-effectively, and do so quickly.

In addition to the extent and speed at which models can consider intervention strategies, models can also consider spatial and temporal scales that are often not feasible in experiments (*Gilligan & van den Bosch, 2008*), or theoretical approaches to disease management that might still be in development. Thus models can provide "outside-the-box" tactics to battle diseases such as HLB. Most experiments cannot logistically test landscape-level disease spread that occurs across multiple years, but this is something that is regularly done with mathematical and statistical models. As an example of "outside-the-box" tactics that models can provide, efforts have been made to control some vector-borne diseases by releasing sterile vectors, which subsequently reduce the vector population and can control or even eliminate the disease (*Thomé, Yang & Esteva, 2010*; *Harris et al., 2012*). Although this is not presently a reality for HLB, models can test

whether this could be an effective control measure for this system and, for instance, provide guidelines on how many sterile psyllids would have to be released to control or eliminate HLB, thus providing insights into the feasibility of this and other hypothetical control strategies.

Here, we provide a case for the more thorough integration of data-driven modeling to HLB control. We first provide case studies of other vector-borne crop disease systems where models were critical to identifying cost-effective management strategies. Next, we discuss previous mathematical models of HLB to reveal how modeling has already advanced study of the HLB system. We then provide an example of how a mathematical model for malaria can be adapted to describe HLB transmission and the potential insights it can yield. In particular, we show how we can identify parameters that require further experimentation or that determine the success of potential intervention strategies.

## MATHEMATICAL MODELS OF VECTOR-BORNE PLANT DISEASES

Plant viruses transmitted by arthropod vectors are a major source of yield losses, infecting a wide range of crop plants. However, the biological details—such as the role of alternative hosts of vectors, the rate of migration, and the seasonality of disease—differ immensely between crop systems and sites. This means that our understanding of disease dynamics and control in one system may not apply to other sites or crops. One way to bridge this control gap is to incorporate biological knowledge into mathematical models that predict disease dynamics and how yield loss will respond to interventions. Models can compare, using a common currency, the potential impact of different interventions by examining sensitivity to parameters that represent different strategies. Previous studies have strategically used models to disentangle the potential role of vector migration, spillover from alternative hosts, and control measures (spraying, netting, phytosanitation) across a range of diseases (e.g., *Fishman et al., 1983*; *Kendall, Brain & Chinn, 1992*; *Holt et al., 1997*, *Holt, Colvin & Muniyappa, 1999*; *Smith et al., 1998*; *Robert, Woodford & Ducray-Bourdin, 2000*; *Zhang, Holt & Colvin, 2001*; *Jeger, 2000*; *Smith & Holt, 1997*). For illustration, we highlight a few key examples here.

*Holt et al. (1997)* describe an African cassava mosaic geminivirus (ACMV) outbreak in cassava, transmitted by a cassava-specific whitefly strain, which was then sweeping through Uganda. The virus also spreads through stem cuttings, the main propagation method for cassava in Africa. Potential control options included phytosanitation (use of uninfected cuttings) and roguing (removal of infected plants). Phytosanitation would be more effective if infected cuttings were driving disease spread, whereas roguing would be more important in a largely vector-driven epidemic. The authors addressed the dynamics and control of this disease using a model that tracks susceptible and infected plants and non-infective and infective vectors, using a version of the Lotka-Volterra predator-prey model with density dependence in both plant and vector. Because there are no alternative vectors or hosts, a minimum density of cassava is required to sustain whitefly populations. The model uncovered otherwise cryptic disease dynamics.

Namely, disease cycles occurred when transmission was only via vectors, whereas when infected cuttings were used in a frequency-dependent manner (i.e., as a low proportion of the total cuttings), disease incidence had a sharp threshold. In this situation, it was difficult to detect when the system was close to a critical transition from low to high disease incidence, causing a collapse of uninfected plants. As a result, crop intensification could increase disease incidence gradually while imperceptibly pushing the system toward collapse. Roguing does not reduce disease incidence but can prevent collapse by pulling the system away from the critical threshold, providing a hidden benefit that would not be detectable without the model.

In a later paper, *Holt, Colvin & Muniyappa (1999)* describe how tomato yields in India suffered massive losses (47–85%) from a whitefly-vectored tomato leaf curl geminivirus (TLCV). In contrast to the cassava example, tomato was only an occasional host for this whitefly, and spillover from other perennials and weedy plants drove vector and virus dynamics. In this context, the authors asked "what is the best method for disease control?" Because most of the vector lifespan occurs on other hosts, the authors adapted a previous general model framework (*Jeger et al., 1998*) to decouple vector dynamics from crop dynamics. The parameterized model could match epidemic curves for susceptible and resistant varieties, although it did not reproduce the 100% prevalence that can occur in fully susceptible populations. Sensitivity analyses were then used to explore different disease management options. Because the tomato crop was a sink for whiteflies and TLCV, interventions that reduce vector immigration and survival were predicted to be most effective. The authors' models suggested that the most effective disease control method would be to distribute netting treated with a persistent insecticide and colored yellow on the crop side; the netting would increase vector mortality and decrease vector immigration and, because the flies are attracted to yellow, the yellow coloring on the crop side would increase emigration. However, because vector migration from uncontrolled populations in alternate plant hosts would sustain the supply of migrants, interventions would need to be continuous to be effective in the long term. Thus, although this system—a whitefly-vectored geminivirus—is superficially similar to the previous cassava example, it highlights the importance of rigorously considering vector, virus, and host biology in a model to design effective interventions. The insecticide-treated, yellow-colored netting devised here is an example of the value of combining complementary approaches to disease control described above, which often only become clear after examining model outcomes.

## MATHEMATICAL MODELS AND HLB

Few mathematical models of HLB currently exist that analyze how HLB spreads within individual trees, within a citrus grove, or from grove to grove. We review here those models which have been applied to HLB because they demonstrate the major insights models have already provided to this disease system. Recent modeling of HLB includes *Jacobsen, Stupiansky & Pilyugin (2013)*, *Parry et al. (2014)* and *Lee et al. (2015)*. These articles elucidate the spread of HLB using three different approaches, namely through differential equation modeling, statistical analysis, and individual-based

modeling, respectively. All of these approaches have benefits and offer insights on different aspects of the system.

*Jacobsen, Stupiansky & Pilyugin (2013)* use a model that is an elaboration on an SIR-type compartment model to understand disease dynamics. Models such as these are among the simplest approaches because they do not necessarily require direct parameterization from experimental data. Nevertheless, they still can provide important insights. *Jacobsen, Stupiansky & Pilyugin (2013)* model the number of trees within a grove that are in four classes: susceptible; infected but not symptomatic; infected and symptomatic; and dead. With their model, *Jacobsen, Stupiansky & Pilyugin (2013)* analyze how the numbers in each class change over time due to bacterial transmission between trees and psyllids. The focus is on what is the range of potential outcomes of disease spread, rather than using a directly parameterized model to make quantitative predictions, i.e. it is a strategic model. However, with the speed of implementing mechanistic modeling and the freedom to consider ranges of solutions, it is possible to find general insights quickly. For example, the elegantly basic model of *Jacobsen, Stupiansky & Pilyugin (2013)* suggested a rather counter-intuitive outcome: if infected trees leave behind infected root stock when rogued that can infect trees newly planted at that location, the best control strategy is actually not to rogue at all. This is because the soil is acting as a reservoir to continue disease spread. However, this relies on the assumption that dead trees do not spread infection which may be false, at least for a short time. Thus, the mathematical model has led to a set of concrete outcomes linked to explicit assumptions, both of which can guide further experimentation.

The work of *Parry et al. (2014)* builds upon the framework of the mechanistic model by fitting a spatially explicit disease model in which trees are either Susceptible, Exposed, Infectious, Detected or Removed using data from Southern Garden's citrus groves. It is primarily a methods paper, using HLB as a case study. Specifically, this modeling entailed estimating parameters from a newly emerged HLB outbreak, that could then be used to predict future disease spread and the impact of control strategies from the early stages of the epidemic. The methods are much more complex, both in terms of mathematics and computational implementation. The available data are discrete snapshots of the disease status of the whole grove—often the case with HLB-infected groves. Using censored detection data with no means to determine the actual exposure and infection time for each tree necessitates specialized statistical methods and bespoke software. Their method is able to determine the transmission process from tree to tree in the presence of psyllid management practices—previous modeling of this sort required the pure disease system without external interference through control. From their modeling, they also determine the effect of tree age on transmission parameters and show that host susceptibility is seasonal, leading to better estimates of parameters for future use. The ability to gain so much information from little data results in better predictions for the continued epidemic and the capability to control the current and future outbreaks. While experiments can be used to calculate estimates for such parameters as infection times or probability of successful transmission, this is not possible when an infection has just emerged. Thus, this modeling allows us to implement control

strategies straightaway without losing our ability to estimate necessary epidemiological parameters to predict the spread of the epidemic.

Finally, *Lee et al. (2015)* combine experiments and individual-based mathematical models. The main experimental result was that, despite being asymptomatic, the host plant can become infectious in a shorter time than previously thought, within 15 days. They used these experimental data in their individual-based model, which describes how the pattern of HLB spread in a grove depends upon the location within the grove that psyllids initially invade. Their model revealed that the average time until a grove is 100% infected is much lower if the psyllids arrive by wind into the center of the grove than if they invade the grove's edge. Thus, if the grower knows that the psyllids were blown in by wind, they should expect that a more intense control strategy is necessary to have any chance of stopping infection. Through mathematical modeling, *Lee et al. (2015)* also found that it is possible for the whole grove to be infected before the first symptoms appear on any tree. From this, they emphasize the need to control psyllid populations regardless of whether any trees have shown symptoms because transmission may already be occurring from asymptomatic trees. Importantly, both of the latter two modeling approaches involved a close integration of the model with biological data to estimate parameters and validate model results. Model-data integration greatly improves the ability of mathematical models to accurately predict best management practices to combat HLB.

In contrast to the three models described above, which explore transmission within a population, *Chiyaka et al. (2012)* focuses instead on disease dynamics within a single tree, due to infection spreading among the different flush patches on the tree. It is one of the first papers to highlight the importance of flush for psyllid dynamics. Flush patches are areas of new leaf shoots on a tree; eggs are deposited on new flush and nymphs remain on the flush during their development. The model is used to determine the role of internal transmission within the tree in comparison to psyllid transmission, and the effect of adult psyllids acquiring transmission as nymphs. Further, they assess the effect of insecticides and removal of infected flush on the proportion of both symptomatic and asymptomatic flush. They find that the role of internal transmission is very important, such that a tree can be 100% symptomatic and die within five years if no insecticides are applied. They also found that removal of flush may not be an effective strategy, depending on initiation time of the control strategy and the frequency of removal.

Additionally, modeling papers exist in which the focus is controlling other citrus diseases. *Cunniffe et al. (2015)* is a good example of a modeling paper that aims to provide useful recommendations to stakeholders such as policy makers and growers, with explanations of why those recommendations are best. The authors include publicly-available software to allow stakeholders to interact with the model, to understand how a strategy of roguing within a radius of detected infected trees would be affected by different roguing radii and the stochastic nature of disease spread. Their focus is on citrus canker but they include HLB as a second example, with the result that optimal roguing radii can be found dependent on the level of risk aversion of the grower. Similarly, *Cunniffe et al. (2014)*, which focuses on Bahia bark scaling of citrus, illustrates that

mathematical models are able to provide useful recommendations for roguing and tree spacing strategies, even when epidemiological knowledge of the disease is limited.

## A PARAMETERIZED HLB MODEL THAT CONSIDERS ECONOMIC COSTS AND BENEFITS

We provide an example of a mathematical model for HLB to illustrate how even simple models can provide useful information for stakeholders, inform laboratory and field experiments, and aid in development of new intervention strategies. We highlight how sensitivity analysis can inform which parameters are lacking in data, thereby suggesting areas for new experimental studies, or which parameters should be targeted for intervention. We demonstrate how to incorporate interventions and economic costs and benefits into a plant disease model and the types of information that models provide. We use a similar mechanistic modeling approach to *Jacobsen, Stupiansky & Pilyugin (2013)* while incorporating realistic parameter values and including data on the temperature dependency of psyllid vital rates. Both *Parry et al. (2014)* and *Lee et al. (2015)* include seasonal aspects within their models, but our model is the first to incorporate the role of temperature on psyllid traits for a wide range of temperature values.

### Model development and assumptions

We adapt a previous model developed by *Parham & Michael (2010)* for malaria, with changes to accommodate psyllid and tree biology and some differences in parameter interpretation. Of particular note, the "biting rate" for mosquitoes will instead be the "feeding rate" for psyllids. The model is parameterized using data from the HLB system (see below). The use of a malaria model highlights the broad applicability of mathematical models that can allow understanding of many vector-borne systems by studying one in detail. The main components of the model are similar to traditional models of vector-borne disease developed by *Ross (1910)* and *Macdonald (1952)*, also for malaria. Citrus trees are categorized as either Susceptible, Asymptomatic or Infected (Fig. 1), in which Infected implies the disease is detectable by symptoms; we assume Asymptomatic and Infected trees transmit the pathogen with the same probability. After an incubation period included in our model through a time delay, asymptomatic trees can transmit infection (*Gottwald, 2010*). Trees develop symptoms over time and move into the Infected class. We include roguing in our model to account for severe symptoms of infected trees resulting in necessary removal of the tree, both to prevent further transmission and due to the lack of profit made from the tree. A very small rate of natural death of susceptible and asymptomatic trees occurs. The sum of trees dying by natural death or roguing equals the total number of trees removed, which are tracked to estimate the costs of roguing. All of these removed trees are assumed to be immediately replaced by susceptible trees in the grove, thus the grove size remains constant. Adult psyllids are Susceptible, Exposed, or Infected, where Exposed indicates that the psyllids are infected but are not yet able to pass the disease on to another tree. The development of eggs and nymphs is included within the birth rate of psyllids. Transmission of infection can occur when an infected psyllid feeds off a susceptible tree, or a susceptible psyllid feeds off an infected

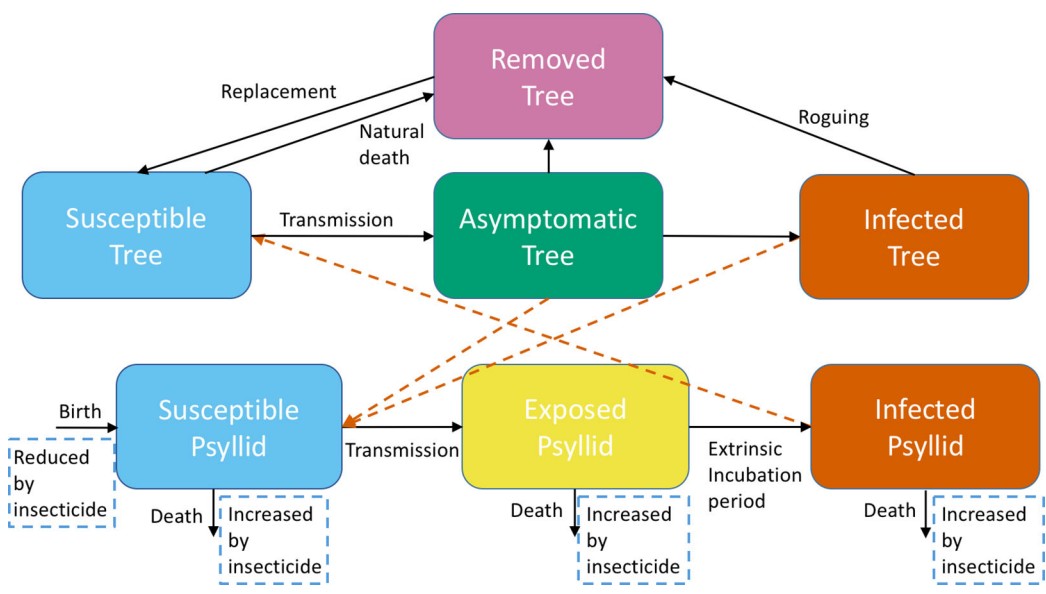

**Figure 1** **A schematic of the model system showing transitions to different categories for trees and adult psyllids.** Trees are either Susceptible, Asymptomatic, Infected or Removed. Adult psyllids are either Susceptible, Exposed or Infected. Black arrows show the transitions between compartments. Orange dashed arrows show the necessary interactions between trees and psyllids to obtain transmission. Light blue dashed boxes highlight how our intervention strategy impacts transitions within the model.

tree. Psyllids have a constant feeding rate which is independent of the number of trees. We assume that the psyllids and trees mix equally with each other—this implies that any psyllid can feed off any tree at any one time. This simplifying assumption is common in vector-borne disease models, but could be modified in future models to reflect the fact that trees are stationary. We assume that the grove has 100% susceptible trees initially (1,001 in total), with psyllids feeding freely. At time 0, we introduce one infected tree. We consider the change in numbers of susceptible, asymptomatic, infected and removed trees for the following 20 years to understand the effects of the initial infection on the whole grove. A full description of the model, with parameter values and information on how we include intervention strategies, is given in Article S1.

## Baseline model parameterization and exploration

An important aspect of our model of HLB is our focus on the seasonality inherent in the psyllid life history. We incorporate the role of temperature on psyllid traits for a wide range of realistic temperature values for Florida. Psyllids are ectotherms and thus will be sensitive to fluctuations in temperature both daily and throughout the year. The thermal physiology of ectotherms has been explored in depth, and it is widely recognized that most traits exhibit unimodal patterns—i.e., performance is low at cold temperatures, ramps up to an optimum, and then falls off as temperature increases further (*Dell, Pawar & Savage, 2011*; *Amarasekare & Savage, 2012*). Recent work on malaria indicates that it is important to incorporate the thermal performance of vectors into disease transmission models (*Mordecai et al., 2013*; *Johnson et al., 2015*). Based on data from *Liu & Tsai (2000)* and average monthly temperatures in Florida, we include yearly variation in

psyllid vital rates, specifically fecundity, development rate, probability of developing from egg to adult, and death rate (see Article S1 and Fig. S1.1).

The birth rate of psyllids is the birth rate per flush patch on a tree. We include seasonal variation in the number of flush patches on a tree, using a similar sinusoidal equation for flush to *Chiyaka et al. (2012)* which produces two flush seasons a year, in spring and autumn (see Article S1).

For our other parameters, such as feeding rate, extrinsic incubation period, and probability of successful transmission between tree and psyllid (and vice versa), we obtained data from a variety of sources including *Pelz-Stelinski et al. (2010)*, *Hall & Albrigo (2007)*, *Gottwald (2010)* and *Martini, Pelz-Stelinski & Stelinski (2015)*. For full details of parameter values and their sources, see Table S1.2.

We build in expected costs, income and profits into our model to assess the impact of disease on the grower and the most cost-effective control strategies. We include the cost of removing a tree and replanting with a new disease-free tree, and the cost of our intervention strategy, namely the cost of one day of insecticide spraying. These costs, as well as the expected profits from susceptible, asymptomatic and infected trees, are estimated from *Stansly et al. (2014)* and *Spreen et al. (2006)*. We assume the profits are constant over time for simplicity (with a discount factor); in reality, profits will change over the course of the outbreak due to changes in supply of citrus (*Florida Agricultural Statistics Service, 2015*). Further details of the inclusion of cost estimates in our model can be found in Article S1.

We first present the model predictions for spread of HLB within a single grove with only roguing. Studying this base case provides the platform for understanding how effective the insecticide intervention strategy is at reducing disease prevalence. Next, we perform a sensitivity analysis to examine which parameters have the most impact on disease dynamics. We then evaluate the effectiveness of a commonly used control strategy, insecticide, at counteracting disease prevalence. We assess the cost-effectiveness of this strategy, which can lead to non-intuitive conclusions about the best strategy to implement.

### Results from the base model with only roguing

With no intervention strategy other than roguing, the disease spreads quickly throughout the grove such that all trees are asymptomatic or infected in less than three years (Fig. 2). Virtually all trees are asymptomatic before there are any infected (symptomatic) trees present, indicating that it is possible for the whole grove to be infected without the grower seeing any symptoms. The progression from asymptomatic to infected trees is slower, reaching a plateau at 2/3's of the grove within eight years. After eight years, the constant replacement of infected trees with new susceptible trees is balanced by new infections, such that the number of asymptomatic and infected trees remains constant over time (see Fig. S2.1). After 20 years, roguing has resulted in replacing over 1,600 trees for a grove size of 1,001 trees—clearly a costly control strategy.

In both summer and winter the temperature in Florida is not well suited for psyllids, which causes clear fluctuations in psyllid population abundance twice each year (Fig. 2B);

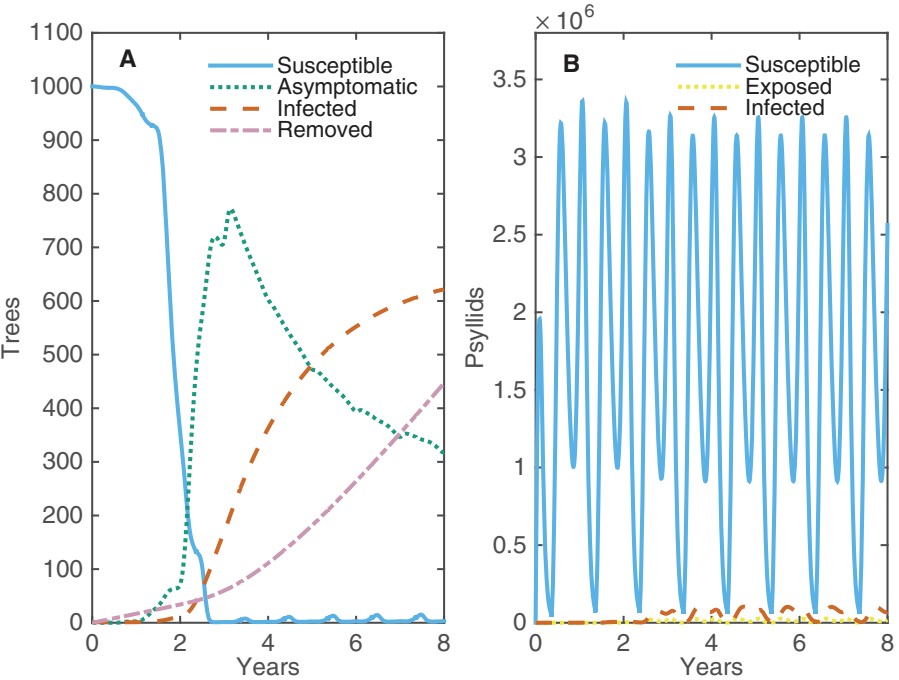

**Figure 2 The changes in numbers of susceptible, asymptomatic and infected trees and susceptible, exposed and infected psyllids over eight years when one tree is infected at time 0.** (A) Alive trees are either Susceptible (light blue), Asymptomatic (green) or Infected (orange), and Removed trees are also plotted (purple). (B) Psyllids are Susceptible (light blue), Exposed (yellow) or Infected (orange).

the bigger dip occurs each summer when temperatures are too high for psyllids to produce eggs. However, their high fecundity the rest of the year allows the psyllid population to bounce back quickly and be effective at spreading disease. A very small proportion of psyllids are exposed or infected at any one time; however, their large population size and success at transmitting the pathogen allows for high levels of prevalence within the grove. Roguing prevents the whole grove from becoming infected, maintaining a balance between the number of infected and asymptomatic trees. This allows for some profits to be made from asymptomatic trees. A potential strategy to reduce the proportion of infected trees in the grove would be roguing trees more quickly after they become infectious. We investigate how changing the average time until a tree is rogued affects both the maximum prevalence and the number of removed trees after 20 years (Fig. S2.2). Roguing trees sooner reduces the peak number of infected trees, but this is outweighed by the significant increase in number of trees that need replaced. However, roguing can have benefits when implemented alongside other control strategies which target different aspects of the disease spread, such as the role of the vector.

## Sensitivity analysis

Through sensitivity analysis we can assess which parameters are influential in the spread of disease within a grove, highlighting which parameters are important to target for intervention or for more experimental study. We focus on $R_0$, the expected number of secondary cases, i.e. the number of trees that will become infected due to a single infected
tree present within a grove (see Article S1). $R_0$ is a combination of parameters related to both the psyllids and the trees, but with a higher proportion of the former. We perform two different types of sensitivity analysis because we have both temperature dependent parameters and constant parameters.

As outlined earlier, we have data on how some psyllid vital rates depend on temperature. In Article S1 and Fig. S1.1, we show response curves fit to those data. These response curves describe how four parameters are affected by temperature: fecundity of female psyllids ($EFD$); the probability of egg to adult survival of psyllids ($p_{EA}$); psyllid development rate ($MDR$); and psyllid death rate ($\mu$). We can also represent the changing numbers of flush patches ($F$) throughout the year as a function of temperature. Performing local sensitivity analysis with these parameters where we vary parameters across a small range, we can assess how changes in temperature propagate through the different parameters to affect $R_0$ (Fig. 3A).

The number of flush patches ($F$) drives the change in $R_0$ as temperature decreases because there are no flush patches at low temperatures. Fecundity also affects $R_0$ at low temperatures but it is most influential at higher temperatures, where low fecundity drives $R_0$ to zero (Fig. 3A). Experimental studies demonstrate that psyllid fecundity is greatly reduced for low and high temperatures. Since $R_0$ is very sensitive to this result, especially at the higher temperatures, it highlights the need to perform more experimental studies of psyllid fecundity for a wide temperature range to ensure its validity. Figure 3A indicates that $\mu$ is influential in reducing $R_0$ at mid to high temperatures, whereas it is not influential at low temperatures. Therefore, an intervention strategy targeting psyllid death rate would be most successful if it is implemented during seasons with intermediate to warm temperatures.

We also perform sensitivity analysis of the constant parameters that are included in $R_0$ (Fig. 3B). For the following parameters we vary its value by 10% and plot the effect on $R_0$: the feeding rate of the psyllid ($a$); the probability of successful transmission from psyllid to tree ($b$); the probability of successful transmission from tree to psyllid ($c$); the roguing rate of infected trees ($r_1$); the natural death rate of susceptible and asymptomatic trees ($r$); the exposure period in trees ($\tau$); the rate of moving from the asymptomatic to the infected class ($\gamma$); and the rate of extrinsic incubation within the psyllid ($\phi$).

The feeding rate of psyllids (parameter $a$, Fig. 3B) clearly has the most effect on $R_0$ of all the constant parameters. This occurs because the parameter is involved in both directions of transmission: from tree to psyllid and vice versa. However, it is hard to experimentally determine the feeding rate of psyllids on trees as they do not follow the pattern of one feed per oviposition, such as mosquitoes, and the nymphs remain attached to tree flush for the duration of this life stage. Thus, gathering more information on this parameter should greatly improve the precision of predictions from this model. Note that parameters $r$, $\gamma$ and $r_1$ have the opposite effect to the other parameters. For example, an increase in $r_1$ decreases $R_0$, whereas other parameters are positively correlated with $R_0$. Apart from $a$, parameters $b$, $c$ (probability of successful transmission) and $r_1$

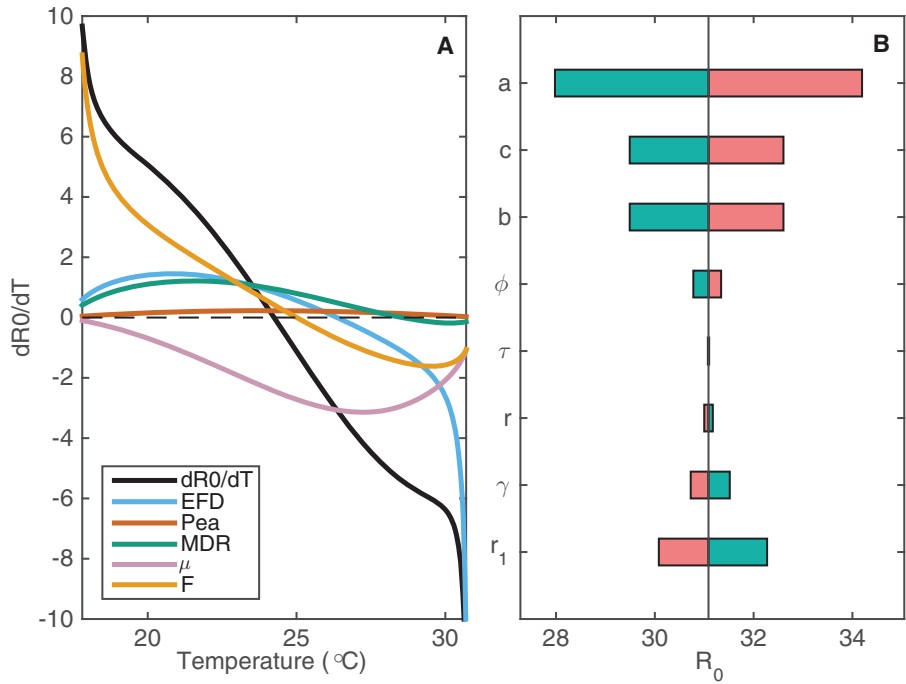

**Figure 3 The sensitivity of $R_0$ to changes in temperature-varying and constant parameters.** (A) The change in $R_0$, $\frac{dR_0}{dT}$, is plotted against temperature (black curve) alongside how each of the psyllid parameters contributes to this change in $R_0$, denoted by color. The contribution by each parameter $p$ on change in $R_0$ is given by $\frac{\partial R_0}{\partial p} \frac{dp}{dT}$. The further the curve is from zero, the more contribution that parameter has to changes in $R_0$. (B) The sensitivity of $R_0$ to changes in constant parameters at $T = 24.25\,°C$, when $R_0$ is at its maximum ($R_0 = 31.09$). Each parameter is varied by 10% to assess the impact on $R_0$ at this maximal temperature. Decreasing each parameter by 10% (e.g. $0.9a$) is indicated in green; increasing each parameter by 10% (e.g. $1.1a$) is indicated in pink.

(roguing rate) have the most effect on $R_0$. By increasing the roguing level it is possible to reduce the prevalence of the disease, but at a high cost (Fig. S2.2).

## Insecticide intervention

Based on the sensitivity analysis, in which psyllid death rate has a significant effect on $R_0$, and since insecticide is currently the main control strategy in use, we analyze what is the most cost-effective strategy to implement insecticide intervention. While many groves are being sprayed with insecticides to control psyllids and thus HLB (*Grafton-Cardwell, Stelinski & Stansly, 2013*), the range of methods for insecticide application across the US is large with differences in number of applications per year and the efficiency of the insecticide (*Qureshi, Kostyk & Stansly, 2014*; *Rogers, 2008*). We assess insecticide application efficiency through a process akin to global sensitivity—we vary the level of insecticide efficiency and the number of days spraying throughout the year over a wide range to capture the current state of play of insecticide application. Our intention here is to present preliminary results and proof-of-concept for the use of a vector-driven epidemiological model to compare the effectiveness of different control scenarios in an isolated grove.

When insecticide is applied to the groves it targets all adult psyllids through increasing their death rate. In our model, it also reduces the birth rate of psyllids, to represent the insecticide killing eggs and nymphs (Fig. 1). We assume that insecticide spray is applied in both spring and autumn. Therefore, insecticide spraying occurs at mid-range temperatures, which our sensitivity analysis indicated was the best time to target the psyllid death rate (Fig. 3A). Spring and autumn sprays involve spraying for the same number of consecutive days. Between simulations we vary the total number of days spraying each year. For example, a simulation with 20 days spraying per year will have 10 consecutive days in the middle of spring and 10 consecutive days in the middle of autumn, whereas a simulation with 10 days spraying per year will have five days each in spring and autumn. Each additional day of spraying costs more money to the grower. We also vary the effectiveness of the spray and assume that it correlated positively with its cost. Sprays that are not very effective cost approximately $15 per day to spray, while highly effective sprays can cost up to $70 per day, for the whole grove. An estimate of $30 per spray is estimated from *Stansly et al. (2014)*. To see full details on how insecticide is included into the mathematical model and how costs of spraying are calculated, see Article S1.

In our model, the number of insecticide application days varies between 10 and 60 days per year split equally between the two spraying sessions, and the efficacy of insecticide applications varies between 60 and 99%. Multiple sprays in a year occur in most groves, with varying ranges of up to seven sprays (*Stansly et al., 2014*), monthly (*Rogers, 2008*), or up to 20 sprays per year (*Spreen et al., 2006*), using a variety of approved sprays that differ in effectiveness (*Rogers, 2008*); some sprays can have an average efficiency as low as 53% (*Qureshi, Kostyk & Stansly, 2014*). Although 60 days is unrealistic logistically in terms of potential insecticide strategies it allows us to investigate the effect of very aggressive control. We quantify how the variation in number of days spraying and effectiveness of spray affect the peak number of infected psyllids and the expected profits from the grove over a 20 year time span (Fig. 4). We also plot a corresponding figure assessing different metrics of insecticide effectiveness, such as peak number of infected trees, in Fig. S2.3.

There is a clear pattern that increasing the number of application days leads to a large reduction in infected psyllids and hence reduced disease spread (Fig. 4A). However, this is not the case for increasing the effectiveness of the insecticide spray. It does lead to reductions in the peak numbers of infected psyllids (the horizontal change in color occurs sooner for highly effective sprays) but the change is slight. Overall, by increasing the effectiveness of the spray and by spraying for more days, the peak number of infected psyllids is lessened from 5.4 to $3.4 \times 10^4$ psyllids. This is a huge reduction in number of infected psyllids but it does not lead to correspondingly large reductions in disease spread. As seen in Fig. S2.3, the peak number of infected trees is effectively unchanged in a grove of size 1,001.

The increasing costs associated with, and the lack of improvement attained through, using more effective sprays, combine to lead to smaller profits as effectiveness increases (Fig. 4B). The additional costs of more effective sprays are not outweighed by the slight reduction in infected trees. In fact, the most cost-effective spraying strategy is

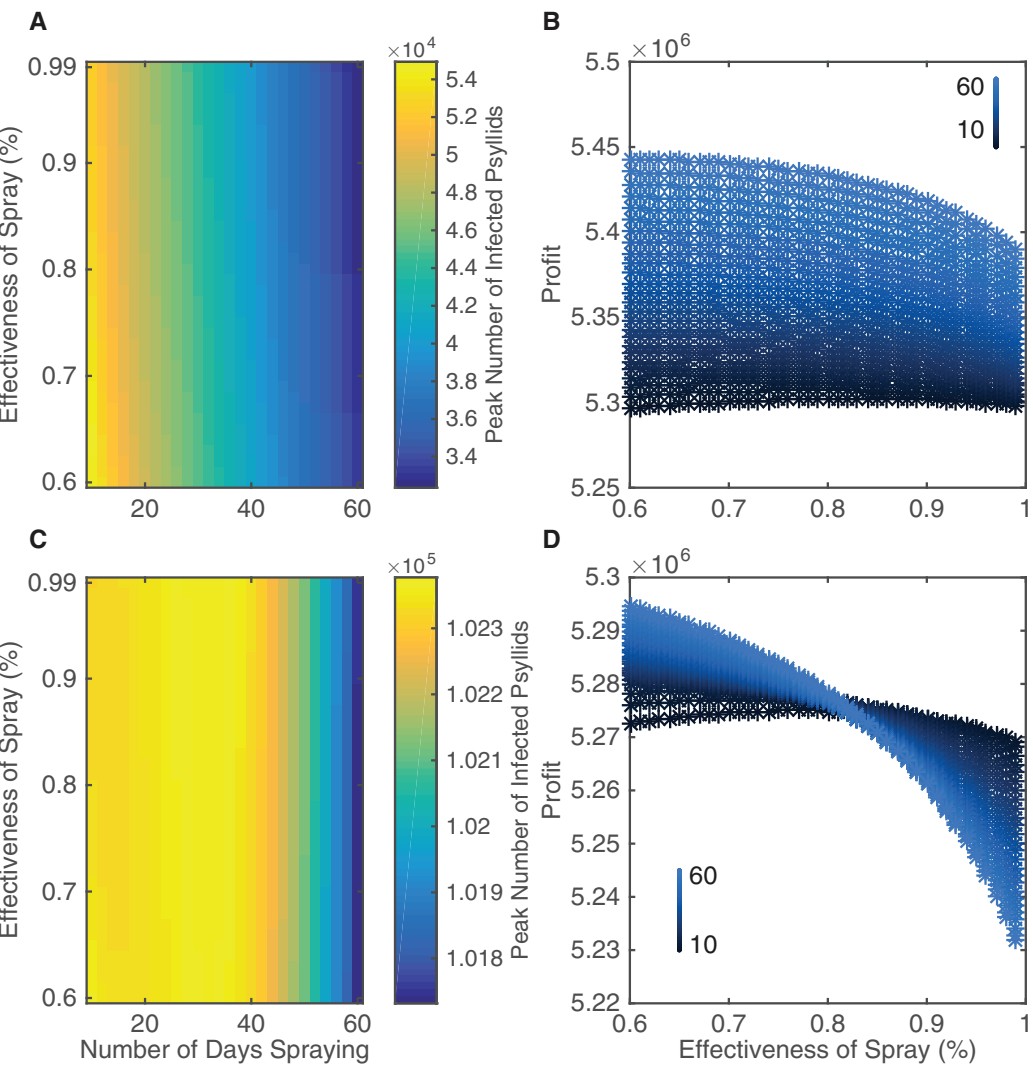

**Figure 4 The effect of different insecticide strategies after a 20 year time span.** Insecticide targets both the death rate and birth rate of psyllids. (A and B) The insecticide is sprayed in spring and autumn. (C and D) The insecticide is sprayed in summer and winter. (A and C) The peak number of infected psyllids is shown as a function of both the number of days spraying during each year and the effectiveness of the spray. (B and D) The end profit after 20 years is plotted against effectiveness of spray when a more effective spray correlates positively with cost. As the points change from black to blue, the number of days spraying per year increases from 10 to 60 days as indicated by the key. The number of days spraying is the total per year, split equally between the two spraying regimes.

60 days at 60% effectiveness. For a wide range of number of spraying days, more profits are gained through choosing the 60% effective spray than a more effective spray.

Figures 4A and 4B highlight that the best strategy is not to search for a more effective spray but to implement the most aggressive control strategy which is logistically possible. With 60% effectiveness, increasing the number of days spraying always led to significant increases in profits. Therefore the limiting factor of extending the number of spraying days is not diminishing returns, but the ability to perform the insecticide application.

For comparison, instead of implementing a spring/autumn spray, as was suggested as the best option by the sensitivity analysis, we consider a summer/winter spraying strategy

(Figs. 4C and 4D). It is clear that spraying in summer and winter is nowhere near as successful as spraying in spring and autumn when one looks at the scale of Fig. 4C. In Fig. 4C, the reduction in peak number of infected psyllids is approximately 500 psyllids, with total number of infected psyllids almost double the highest number in Fig. 4A. Thus, the best intervention when spraying in summer and winter is much worse than the worst intervention when spraying in spring and autumn, in terms of number of infected psyllids. This propagates into profits as well, with lower profits achieved for spraying in summer and winter (Fig. 4D). This highlights the importance of considering the seasonality inherent in the system, as it will affect when to implement intervention strategies. In Fig. 4D, the effectiveness of the spray is of so little value in reducing psyllid numbers that profits actually decrease when we spray for more days with a costlier, more effective spray. This occurs because increasing spraying days only reduces infection prevalence very little, and therefore there is no significant increase in income to outweigh the extra costs of spraying. Thus, our results validate our sensitivity analysis which indicated that the death rate had most effect on the spread of the disease in mid-range temperatures.

We present the results for the expected citrus profits when there is HLB and 1) no intervention, 2) insecticide application, and the ideal but currently unrealistic scenario of 3) no HLB (Table 1). Costs are included in the model as outlined in Article S1. We focus on the most successful intervention strategy presented, which was spraying for 60 days in spring and autumn, with 60% effectiveness.

The large cost of insecticide application is outweighed by the increase in income compared with the no intervention case (Table 1). Thus, insecticide application looks promising. By viewing the profits over a 20 year time frame (Fig. 5), it is possible to gain more understanding of how insecticide profits compare with the other scenarios. Both insecticide application and the no intervention strategy manage to maintain similar profits to the no disease case for the first five years. Around this time, the proportion of infected trees becomes larger than the proportion of asymptomatic trees, hindering profits significantly (see Fig. 2). The insecticide strategy maintains higher profits than not intervening but the improvement is not large. This suggests that while insecticide application is useful for maintaining profits, it is not going to boost profits in HLB-infected groves in the long run.

## Model summary

We have used a previously existing malaria model and adapted it to HLB by adding in temperature-dependent parameters for psyllid vital rates, roguing of trees, flushing of trees and economic costs. This model is clearly preliminary and only a first step towards understanding the spread of HLB within a grove, with a more HLB-specific model required to be able to capture the full dynamics of the citrus, psyllid and pathogen interactions. However, the relatively simple model presented here, that captures the main features of HLB spread, is able to establish useful recommendations for managing HLB.

Using sensitivity analysis, we are able to suggest what new data need to be collected, or which parameters to focus on for potential intervention strategies. In particular, our

**Table 1 The expected costs and income for different intervention scenarios, rounded to the nearest dollar.** The insecticide treatment is 60% effective, with 60 days of spraying (Fig. 4B). The no intervention strategy includes roguing of infected trees, as in Fig. 2. The no disease case includes natural death and replacement of susceptible trees. All other parameters are as in Tables S1.2 and S1.3.

|  | No disease | No intervention (Fig. 2) | Insecticide (Fig. 4B) |
|---|---|---|---|
| Cost of removing trees | $11,447 | $56,234 | $51,327 |
| Cost of intervention | – | – | $17,644 |
| Income from trees | $7,613,948 | $5,253,263 | $5,511,907 |
| Total profit | $7,602,501 | $5,197,029 | $5,442,936 |

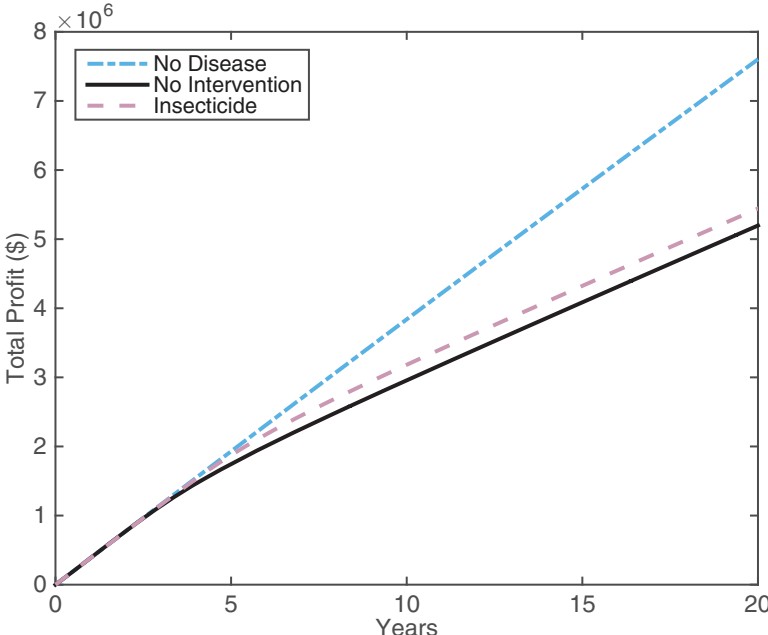

**Figure 5 The profit attained by growers over 20 years for different disease and intervention scenarios: the no disease case (blue dot-dash line), the no intervention case (black solid line), and insecticide spraying (pink dashed line).** The insecticide treatment is 60% effective, with 60 days of spraying (Fig. 4B). All other parameters are in Tables S1.2 and S1.3.

preliminary analyses suggest fecundity of psyllids should be measured over a finer temperature range to better pin down the temperature window for psyllid egg production, since transmission of infection is strongly affected by when psyllids can produce eggs. Similarly, we suggest the feeding rate of psyllids requires further experimental work because it is an important parameter but the amount of data collected for it so far is small. It could also be dependent on temperature which has not been considered in experimental studies. Often the feeding rate is only inferred from studies assessing success or failure of transmission of HLB between tree and psyllid. But this parameter should be independent of whether transmission occurs. Furthermore, through sensitivity analysis, we propose that the death rate of psyllids, especially during mid temperatures, is influential on the spread of the disease and should be targeted for intervention. Performing similar anaylses with an HLB-specific model and more data to parameterize

it will allow us to shortlist the intervention strategies we examine, at least in the initial stages.

Targeting the psyllid death rate through the use of insecticide spray led to a reduction in the disease spread within a grove and increased profits compared with no intervention. One important aspect found was the need to include psyllid temperature dependence and seasonal temperature. This plays a prominent role in the success of intervention strategies, with some times of year much better for reducing infection. However, the degree to which disease is reduced is low even in the best simulations we found. Disease spreads rapidly throughout the grove and the interventions are only capable of maintaining profits rather than eradicating HLB.

As stated above, a more HLB-specific model that captures additional factors involved in HLB spread would produce more reliable and more concrete results for implementing management practices for HLB. To achieve this, the most important update to the model would be to consider the adult and nymphal stages of the psyllids separately. In our model, it is assumed that only adult psyllids are able to be infected and are infectious to trees. In reality, the nymphal stage of psyllids has a significant role to play in the transmission of HLB. Experiments have concluded that most psyllids become infected with HLB when they are nymphs and then remain infected for their entire lifespan (*Pelz-Stelinski et al., 2010*; *Hung et al., 2004*). Psyllids that become infectious as adults often transmit the pathogen to trees at a lower rate than adults who became infected as nymphs (*Pelz-Stelinski et al., 2010*; *Inoue et al., 2009*). However, nymphs usually remain attached to one tree flush in the early nymphal stages and thus do not transmit the disease to other trees (*Hall et al., 2013*). There is also a slim chance of vertical (transovarial) transmission (*Pelz-Stelinski et al., 2010*). Adding the egg and nymph life-stages directly to the model and incorporating the details of how transmission of HLB from psyllid to tree is affected by the psyllid life history would allow us to evaluate the importance of these aspects to the transmission of HLB and the impacts on potential control strategies. Other additions could include a more detailed description of the seasonal growth of flush, non-homogeneous mixing of psyllids and trees, and invasions of psyllids from outside the grove.

## CONCLUSIONS

Collaborations between empiricists and mathematical modelers have the potential to identify solutions to HLB efficiently and reliably. In other plant disease systems, by incorporating the wealth of knowledge provided by empiricists, models have been proven to disentangle the potential drivers of the disease, inform which aspects of the system to target to control disease and the potential efficiency of those intervention strategies. This success can also transpire for HLB, allowing profits to be maintained and the possibility of disease eradication. We have shown that even simple models for HLB can provide useful recommendations for moving forward with disease management. By collaborating more closely with empiricists, these recommendations will improve in scope, reliability and accuracy. Models can highlight our lack of understanding in crucial areas, directing future lab and field work. For example, our model demonstrated that the feeding rate of psyllids is an important component of disease spread. Therefore, better

communication between modelers and empiricists is required, benefiting both groups through improved data collection and models.

We highlight here the tools that mathematical models can bring to the table for fighting HLB. For simpler models, the strength lies in the sensitivity analysis, which allows models to be improved by suggesting better data collection. For future models, perhaps most useful of all is the ability to test different interventions and combinations of strategies in a short time frame to predict which will be the most successful. Improvements can be made to our model to include more aspects of psyllid and tree biology and different intervention strategies can be considered relatively quickly. Other adaptations could be introduced to consider multiple groves, as well as introducing uncertainties in the host response, pathogen and vector dynamics. This reduces the amount of time required performing field experiments to determine if the interventions could work. Furthermore, the ability for economic considerations to be integrated into mathematical models to allow for optimal management of the intervention is a strength that can not be rivaled by other methods. Decisions for future management and control can be made based upon informed analysis of the costs and benefits involved rather than intuition. Therefore, we believe that mathematical models are a powerful method that need to be utilized further for managing the spread of HLB.

### Funding
E.A.M., J.R.R. and L.R.J. were supported by the National Science Foundation (DEB-1518681). The funders had no role in study design, data collection and analysis, decision to publish, or preparation of the manuscript.

### Grant Disclosures
The following grant information was disclosed by the authors:
National Science Foundation: DEB-1518681.

### Competing Interests
The authors declare that they have no competing interests.

### Author Contributions
- Rachel A. Taylor conceived and designed the experiments, performed the experiments, analyzed the data, wrote the paper, prepared figures and/or tables, reviewed drafts of the paper.
- Erin A. Mordecai wrote the paper, reviewed drafts of the paper.
- Christopher A. Gilligan wrote the paper, reviewed drafts of the paper.
- Jason R. Rohr conceived and designed the experiments, wrote the paper, reviewed drafts of the paper.
- Leah R. Johnson conceived and designed the experiments, wrote the paper, reviewed drafts of the paper.

## Data Deposition

Open repository Huanglongbing: https://rataylor@bitbucket.org/rataylor/huanglongbing.git.

## Supplemental Information

Supplemental information for this article can be found online at http://dx.doi.org/10.7717/peerj.2642#supplemental-information.

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
