# Peer review of "Mathematical models are a powerful method to understand and control the spread of Huanglongbing"

_PeerJ, doi:10.7717/peerj.2642_

## Round 0.1 · original submission · Major Revisions

Both reviewers believe your manuscript has great potential, but both raise substantial concerns with the model formulation. Reviewer 1 raises concerns about whether the model captures important features of the disease. Reviewer 2 suggests even more serious problems with the model -- that the model is essentially not consistent. Any revision would need to carefully address these concerns (as well as the other issues raised).

·

Basic reporting

Overall the quality of writing is quite good, the paper is very readable, and the literature is properly referenced. The few exceptions to this are presented below.
There is a paragraph on page 3 of the supplemental material just under the equation for R_0 in which vectors and hosts seem to be reversed. For instance, a line refers to the “number of successful bites from host to vector.”

The 2012 paper of Chiayaka et al. "Modeling huanglongbing transmission within a citrus tree" may deserve some recognition in your review of mathematical models for HLB. Although it is quite different than the other models mentioned since it considers the spread within a tree rather than within the grove, it is notable for being the first (to my knowledge) model to highlight the importance of flush to transmission. Our 2015 paper, reinforced by Bill Dawson’s experimentation, focused on how flush contributes to the spread of HLB within a grove. Even more recent research (Hall et al. 2016) argues that flush is an essential factor of HLB spread.

On line 277, the paper says that attention to seasonality is a highlight of the model. The Parry et al. paper you reference includes some seasonal effects, and even our paper varied some vector and host parameters based on season. It is done much more carefully in your model, but the paper as it currently stands makes it sound like previous papers ignored seasonality entirely.

Experimental design

I appreciate your paper’s focus on the need for data-driven strategic models (line 90 or so) which are both mechanistic and predictive. A compartmental model is not an ideal candidate for such a model. The well-mixing assumptions inherent in such models are not suitable for HLB. Aside from theoretical problems such as hosts being stationary, data shows that symptomatic trees are not distributed spatially uniformly in a grove but rather tend to be found at the edges (for example Gottwald et al. (2008) “The Plantation Edge Effect of HLB – A Geostatistical Analysis”). Despite this, I do believe that it is worthwhile to extract as much information as possible from a compartmental model.

A key feature of this paper is the adaptation of a malaria model to HLB by adjusting parameter values of the malaria model. I am concerned that some underlying assumptions regarding malaria and their resulting effect on the model do not port well to HLB. To start with, the well-mixing assumption of a compartmental model is more likely to be satisfied when both host and vector are moving around. There are other, more important issues, however.

First, the birth rate of psyllids depends only on temperature in the model whereas typically the growth of a population depends on the size of the population and capacity of the environment as well. A constant birth rate suggests that potential oviposition sites (flush shoots) are fully saturated at the expected populations. While this might be true in the case of complete infestation, it will not be true when the population is small relative to the number of hosts. Judging from Figure 2 on page 8, this happens every year even without insecticide.

Second, the model does not consider invasion from outside of the grove. While the migration patterns of psyllids are not well-studied in the scientific literature, they seem to be a significant factor in disease dynamics. In Florida, nearby growers coordinate their sprays primarily to decrease the effects of invasion. They claim that this is effective and have presented data to this extent at conferences. Unfortunately, the results of these Coordinated Health Management Areas (CHMAs) have not been published.

Asymptomatic spread requires more consideration. Parry et al. argues that “it is clear an epidemic can be underway well before symptoms are first seen.” As you note, our paper suggests that an entire grove may have been exposed to HLB before a single tree shows symptoms. Since cryptic/asymptomatic spread is a primary driver of disease dynamics, it should be given some attention. While the previous two issues would require only minor modifications to the malaria model, this change is more significant.

Finally, the central role of flush in both the lifecycle of the psyllid and the transmission of HLB needs to be present in the model.

I understand that the goal of your paper is to have a simple streamlined model, but all of the above are major factors in the dynamics of the disease spread that should be included. Aside from these, I have a few more focused suggestions.

Your parameter table lists our paper as a reference for the tree to psyllid transmission probability. Our model ignored this transmission. Following a paper of Hall et al. (2012) we only allowed nymphs to become infected and mature into infected adults since psyllids who acquire HLB as adults are poor transmitters.

While you changed the parameter “biting rate” in the Malaria model to “feeding rate”, there are still references to “parasites” in the parameter table. I would suggest changing these names as well.

Validity of the findings

The validity of the results hinges on how well the model captures the dynamics of HLB. The analysis of the model as given seems correct and thorough, but due to my reservations, I do not think that the suggestions of the model carry much force.

Additional comments

I feel that this has the potential to be a fantastic paper. It is staggering that nobody has analyzed an elementary compartmental model for HLB. Despite their limitations, doing so would provide a great foundation for more complicated models that incorporate the growing scientific knowledge about HLB, and, as eloquently argued in the paper, would inform scientists of where their experimental efforts are best allocated. However, I feel that at present the malaria model has not been properly adapted to HLB to be that starting point. It is difficult to argue exactly what is essential in order to capture the disease dynamics of HLB, but I strongly feel that incorporating these changes will greatly strengthen the findings.

·

Basic reporting

The article discusses the role of mathematical modelling in the design of pest management strategies and focuses on understanding and mitigating the spread of citrus greening, a plant pathogen. The paper is well written and argued. The authors present a delay differential equation model of the host (citrus trees) and the disease vector (psyllid). The inclusion of temperature dependent rates into their model is novel and often omitted from pest control models. The authors demonstrate that the inclusion of climatic effects gives valuable insights into the optimal timing of control within a season and this links to the biology of the disease vector, whose life-cycle is temperature dependent.

The article is self-contained and the figures are clear and well presented.

There was some aspects of figure 3 A I found confusing which it I think need some clarifying. Firstly, the graph refers the contribution of each parameter to dR0/DT it is a little unclear what this means. Equation S1.12 is being used, but I don't see what exactly is being plotted when for example we look at the mu line, are all the terms involving mu, including dr0/du du/dT being plotted? . I think this needs some clarification especially in the main text. It is a little bit clearer on S1 page 6, but not in the Figure legend. Similarly the figure legend implies that the R0 line is the contribution of R0 to the dR0/dT which I am not sure how to interpret, but the text on S1 page suggests the the line labelled R0 is infact dR0/DT plotted as function of temperature. I think a bit more clarification would be very helpful. Other than this all the figures and explanation are very clear.

Experimental design

Here I am interpreting experimental design as "Model design". The general SIR framework that the authors use is well suited to the plant-pathogen problem, however there are details in the model formulation which I have concerns about. The authors introduce delays into their model to account for the time spent in transition from susceptible to infective tree, this is a reasonable thing to do, but one has to be quite careful with how this is done, particularly when some of the parameters are varying with time, i.e. temperature dependent rates. I will outline the problems below, I think they need to be addressed as they effect terms involving mu and a, both parameters that the authors point out that the model is sensitive too.

(1) Consider the case when the parameters are constant and don't vary with time.

(a) The authors introduce a delay in (S1.2) to describe susceptible trees becoming infective. As the authors rightly note on page 3 of appendix S1 their model assumes no natural death of exposed trees. I appreciate the death rate of the trees is small, but if included it would mean a exp (-r tau)=0.9917 term would multiply the a in equation S1.2 This is quite a small reduction in a, however given that the model is sensitive to a (Fig 3b) it seems this could well have an impact. At the very least I think more discussion is needed to explain why this assumption does not effect results as it essentially allows exposed trees to escape death for a period of tau time units.

(b) In equations S1.18 and S1.19 the authors account for the natural mortality of exposed vectors via the term exp(-mu/phi), however when the control measure is introduced this term should be changed to. It should become exp(-(mu+ins)/phi) . At the moment the model formulation means that during the incubation period psyllid's are not dying from the control, so it's like a protected state which I don't think is correct. As with point 1(a) above, as the results are sensitive to mu and this correction will impact on a mu term the omission could affect results.

(2) Consider the case when the parameters vary with temperature, in other words vary with time.

In this scenario there is a bigger problem with the model formulation. One has to be quite careful with the book keeping in delay differential equation models.
So good references on this are:
Nisbet, R. M. and W. S. C. Gurney (1983). The systematic formulation of population models for insects with dynamically varying instar duration. Theoretical Population Biology 23, 114–135.
and
Molnar, PK, et al (2013) Metabolic approaches to understanding climate change
impacts on seasonal host-macroparasite dynamics. Ecology Letters 16: 9-21

Let's take the simple example of the mu(t) term, so death depends on temperature (and therefore time). Now at the moment S1.18 and S1.19 have surival of the exposed period as exp(mu(t)/phi) so they base the mortality in the exposed period on the current temperature, where as in fact that mortality has been varying over the exposed period.

Let S(t) be the survival during the exposed period, then S(t) satisfies the following ODE:
dS/dt = -mu(t) S
If mu(t) is fixed solving this over the exposed period give exactly the quantity the authors use: exp(-mu/phi). However now mu(t) is temperature and so time dependent, which means the solution is now S(t)=exp( integral mu(t) ) , where the integral is over the exposed time period.
Similarly the issue in 1(b) requires addressing in a similar way.

(Also see the appendix of our paper for a derivation:
Ewing, D.A.; Cobbold, C.A.; Purse, B.V.; Nunn, M.A.; White, S.M.. 2016 Modelling the effect of temperature on the seasonal population dynamics of temperate mosquitoes. Journal of Theoretical Biology, 400. 65-79. 10.1016/j.jtbi.2016.04.008 )

Again given that the model is sensitive to mu then this could change the model output, especially during exposed periods that cover temperature ranges that include temperature extremes where there could be a large variation in mu over the exposed period.

So I think these issues with the model formulation need addressing.

Validity of the findings

For the model presented the results are robust and appropriate conclusions are reached and well evidenced. However as mentioned in Experimental design I have some reservations about the model itself.

Additional comments

My overall impression of the paper is very positive. It's well written and addresses and interesting topic. My only reservation is the model itself and the delay terms that I have mentioned. I think this can be addressed, but I do think it is quite important and I am feel cautious about interpreting the results until these issues are addressed.

---

## Round 0.2 · accepted · Accept

You have done a nice job on the revision.